# Resistance to BRAF Inhibitors: EZH2 and Its Downstream Targets as Potential Therapeutic Options in Melanoma

**DOI:** 10.3390/ijms24031963

**Published:** 2023-01-19

**Authors:** Anne Uebel, Stefanie Kewitz-Hempel, Edith Willscher, Kathleen Gebhardt, Cord Sunderkötter, Dennis Gerloff

**Affiliations:** 1Department of Dermatology and Venereology, University Hospital Halle (Saale), Martin-Luther-University Halle-Wittenberg, 06120 Halle (Saale), Germany; 2Department of Internal Medicine IV, Oncology/Hematology, Martin-Luther-University Halle-Wittenberg, 06120 Halle (Saale), Germany

**Keywords:** melanoma, BRAF, resistance, EZH2, vemurafenib

## Abstract

Activating BRAF mutations occurs in 50–60% of malignant melanomas. Although initially treatable, the development of resistance to BRAF-targeted therapies (BRAFi) is a major challenge and limits their efficacy. We have previously shown that the BRAF_V600E_ signaling pathway mediates the expression of EZH2, an epigenetic regulator related to melanoma progression and worse overall survival. Therefore, we wondered whether inhibition of EZH2 would be a way to overcome resistance to vemurafenib. We found that the addition of an EZH2 inhibitor to vemurafenib improved the response of melanoma cells resistant to BRAFi with regard to decreased viability, cell-cycle arrest and increased apoptosis. By next-generation sequencing, we revealed that the combined inhibition of BRAF and EZH2 dramatically suppresses pathways of mitosis and cell cycle. This effect was linked to the downregulation of Polo-kinase 1 (PLK1), a key regulator of cell cycle and proliferation. Subsequently, when we inhibited PLK1, we found decreased cell viability of melanoma cells resistant to BRAFi. When we inhibited both BRAF and PLK1, we achieved an improved response of BRAFi-resistant melanoma cells, which was comparable to the combined inhibition of BRAF and EZH2. These results thus reveal that targeting EZH2 or its downstream targets, such as PLK1, in combination with BRAF inhibitors are potential novel therapeutic options in melanomas with BRAF mutations.

## 1. Introduction

Malignant melanoma is one of the most aggressive tumors with an increasing incidence [1]. It is associated with a high variety of somatic gene alterations, which affect different cellular processes, such as proliferation, apoptosis or cell-cycle control. Besides other mutations, such as those of NRAS, p53 and PTEN, activating BRAF mutations are most frequently (50–60%) found in malignant melanomas [2,3,4], leading to the constitutively activation of RAF-MEK-MAPK signaling, which leads to increased cell proliferation and melanoma progression [2,5].

Targeted therapies have been established in the last decade aiming to inhibit constitutively activated MAPK. The most effective agents are the BRAF inhibitors vemurafenib and dabrafenib, as well as the MEK inhibitors trametinib and cobimetinib, which interfere in the upstream signaling of MAPK [5]. These targeted therapies improve patients overall survival and disease-free survival [6,7,8,9,10], but about 15% of patients show no response to BRAF inhibition and about 50% of primary responders develop resistance after 6–8 months [11]. The addition of an MEK inhibitor to BRAF inhibition improves clinical outcomes [12,13,14], but the effect of this combination is also limited by primary and acquired resistance, which is still a major challenge.

There are different ways to acquire resistance to BRAF inhibitors. In most cases, MAPK is reactivated by additional mutations, most commonly by NRAS mutations [15,16,17,18], with NRAS located upstream of BRAF. Alterations of MEK also reactivates MAPK signaling pathways [18]. Further mechanisms consist of bypass pathways, such as PI3K/PTEN/AKT signaling [16], alternative splice variations of BRAF-V600 [17,19] and amplifications of BRAF_V600E/K_ [17,20]. In some samples, the co-occurrence of different resistance mechanisms was detected [17]. The emergence of resistance to MEK inhibitors occurs through comparable mechanisms, such as MAPK reactivation or PI3K/PTEN/AKT signaling [21]. For a substantial proportion (approximately 40%) of melanomas that are resistant to targeted therapies, the mechanisms are not yet fully understood. [17,22].

Because epigenetic changes have been shown to contribute to melanoma progression [23], histone methyltransferases and histone deacetylases are also thought to be critical mediators of resistance to targeted therapies. Thus, epigenetic silencing of PP2A has been shown to confer resistance to HER2-targeted therapies in breast cancer [24].

The epigenetic modulator enhancer of zeste homolog 2 (EZH2) is a catalytic subunit of polycomb repressive complex 2 (PRC2) mediating trimethylation of lysine 27 on histone 3 (H3K27me3). This leads to the formation of heterochromatin, resulting in gene silencing and transcriptional inactivation [25,26,27,28]. In a non-canonical way, EZH2 mediates, dependent on PRC2, the methylation of proteins, such as GATA4 or RORα [29,30,31]. Alternatively, it interacts directly, PRC2 independent, with several proteins, such as STAT3 [31,32]. Thus, depending on the interaction, EZH2 modulates both the activation and repression of gene expression. [25,31]. Therefore, EZH2 is involved in the regulation of many important cellular processes such as the cell cycle, proliferation and apoptosis [33,34,35]. In cancer, EZH2 induces epithelial–mesenchymal transition (EMT), which is relevant in metastasis [27,36,37]. Previous research has already shown that EZH2 contributes to the progression of various malignancies, e.g., of the endometrium and breast [38], as well as malignancies of the prostate [38,39] or ovaries [40]. Finally, a high expression of EZH2 is associated with metastasis, aggressive tumors and poor prognosis [33,38,41].

With regard to melanocytic tumors, the expression of EZH2 is higher in malignant melanoma entities than in benign nevi [42]. The BRAF_V600E_ mutation is linked to a high expression of EZH2 [43], which is associated with melanoma progression and worse patient overall survival [33,38,43,44].

EZH2 has been shown to also contribute to the emergence of resistance in various tumors, such as in small-cell lung cancer, head-and-neck squamous cell carcinoma, glioblastoma and ovarian carcinoma [40,45,46,47].

In our previous work, we noticed decreasing levels of EZH2 in BRAF_V600E_-mutated melanoma cells treated with vemurafenib, while cells resistant to BRAF inhibitors showed an unchanged expression of EZH2 upon treatment with vemurafenib [48]. Therefore, we wondered whether targeting EZH2 might represent a new therapeutic option in melanomas resistant to BRAF inhibitors. We found that combined treatment with inhibitors against BRAF and EZH2 indeed showed synergistic effects against cancer cells and improved the response of melanoma cells resistant to vemurafenib. These effects were linked to the suppression of the proto-oncogene PLK1, which is downstream of EZH2.

## 2. Results

### 2.1. Inhibition of EZH2 Improves Response to Vemurafenib in Resistant Cell Lines

Because we found that EZH2 contributes to the emergence of resistance to vemurafenib, we first investigated whether the inhibition of EZH2 by tazemetostat (EPZ-6438; (EPZ)) improves the response to vemurafenib in susceptible cells (A375) and in cells resistant to vemurafenib (A375R). In Western blot analyses, we found a strong decrease in EZH2 expression in susceptible A375 cells after treatment with vemurafenib or a combination of vemurafenib with EPZ. A375R cells also showed decreased EZH2 expression, but the effect was much weaker (Figure 1A). H3K27 trimethylation, a sign of functional EZH2, was greatly reduced in both A375 and A375R cells after combined treatment with vemurafenib and EPZ. In addition, we observed decreased phosphorylation of ERK 1/2 (pERK 1/2) in both A375 and A375R cells after treatment with vemurafenib alone or in combination with EPZ. However, phosphorylation of ERK 1/2 was even more reduced in A375R cells treated with both vemurafenib and EPZ. This implicates an additive effect of EPZ on vemurafenib. In viability assays, A375R cells showed a better response (IC_50Vem_~13 µM) when EPZ was added to treatment with vemurafenib than to vemurafenib alone (IC_50Vem_~22 µM) (Figure 1C), while vemurafenib and EPZ had no additional effect on susceptible A375 cells compared to treatment with vemurafenib alone (Figure 1B).

We obtained comparable results with further melanoma cell lines resistant to vemurafenib (WM9R and WM35R) (Appendix A). There was even a dose–response relationship because cell viability was more reduced by increasing EPZ concentrations (Appendix A).

In viability assays, we found that A375 and A375R cells showed similar tolerance to EPZ (IC_50EPZ_~70 µM) (Figure 1D). We did not detect cytotoxic effects of EPZ on normal melanocytes and normal keratinocytes (Figure 1E). Our results reveal that EPZ has a synergistic effect with vemurafenib on A375R cells, leading to reduced cell viability.

### 2.2. Knockdown of EZH2 Improves Response to Vemurafenib of A375R Cells

To confirm that the enhanced response of A375R cells was induced by the specific inhibition of EZH2, we generated EZH2 knockout cells. For this purpose, A375 and A375R cells were transfected with EZH2-specific shRNA constructs or control shRNAs and selected with puromycin. By Western blot analyses, we validated a strong knockdown for EZH2, as well as decreased trimethylation of H3K27 in A375 cells and A375R cells (Figure 2A).

In susceptible A375 cells treated with vemurafenib, the knockdown of EZH2 had no additional effect on viability compared with the control shRNA (sh-luc). In contrast, A375R cells with EZH2 knockdown showed an improved response to vemurafenib, resulting in decreased cell viability. IC_50_ values of vemurafenib decreased to 36% in comparison with A375R with EZH2 knockdown (sh-EZH2; IC_50Vem_ = 12 µM) and A375R control (sh-luc; IC_50Vem_ = 33 µM).

Thus, these results are comparable to effects seen after the inhibition of EZH2 by EPZ.

The knockdown of EZH2 also led to significantly reduced cell proliferation of A375R cells (Figure 2E), whereas A375 showed only weakly reduced proliferation (Figure 2D).

In summary, these results confirm our previous data and show that the blockade of EZH2 leads to a better response of vemurafenib-resistant cells (A375R).

### 2.3. Combination of Vemurafenib and EPZ Enhances G0/G1 Phase Arrest and Apoptosis in A375R Cells

To analyze how combined treatment with vemurafenib and EPZ affects melanoma cells, we examined cell-cycle distribution and apoptosis. For this purpose, A375 and A375R cells were treated for 72 h with DMSO (control), vemurafenib (A375: 1 µM; A375R: 10 µM) or EPZ (5 µM), or with a combination of vemurafenib and EPZ. In cell-cycle analyses, we found a significant shift in the cell-cycle distribution of A375R cells when EPZ was added to treatment with vemurafenib compared to all other conditions (Figure 3A). When cells were treated with both agents, we observed a large increase in cells remaining in the subG1 and G0/G1 phases. In contrast, for A375, we observed a comparable increase in the G0/G1 phase in cells when treated with vemurafenib alone or in combination with EPZ. Treatment with EPZ alone showed no effect on cell-cycle distribution compared to the control (Appendix A). To investigate these results in more detail, we performed cell-cycle analyses after serum starvation. Here, we observed the following corresponding results: compared to all other conditions, A375R cells treated with EPZ in addition to vemurafenib had the highest number of cells arrested in the G0/G1 phase (non-cyclic cells) after being maintained in a serum-containing medium for 24 h (Figure 3B). A375 cells showed only a small increase in non-cycling cells after treatment with vemurafenib alone or in combination with EPZ (Appendix A).

With regard to apoptosis, a significantly large increase in apoptotic cells for A375R cells was found after combined treatment with vemurafenib and EPZ (Figure 3C), while A375 cells showed only a small increase in apoptotic cells in response to vemurafenib alone or in combination with EPZ (Appendix A).

In conclusion, our results show that the addition of EPZ to vemurafenib increases the percentage of cells which remain in the subG1 and G0/G1 phases, thus reducing the number of cycling cells and increasing the apoptosis of A375R cells.

### 2.4. The Combined Treatment of A375R Cells with Vemurafenib and EPZ Results in a Significant Change in the Expression of Cell-Cycle-Associated Genes

Assuming that the observed changes in viability, cell cycle and apoptosis after the addition of EPZ to vemurafenib go along with the marked effects on gene expression, we performed next-generation sequencing (NGS) of A375R obtained under the above-mentioned conditions. The expression profiles of the control A375R cells (DMSO) resembled those of A375R cells treated with EPZ. A375R cells treated with vemurafenib alone showed greater differences in gene expression (Figure 4A,B), but cells treated with vemurafenib and EPZ showed the greatest distance in gene expression profiles from the other samples, as observed in the principal component analysis (PCA) and hierarchical clustering (Figure 4A,B). When analyzing the differentially expressed genes of A375R cells treated with vemurafenib and EPZ in combination, compared with all other conditions, we found an overlap of 5043 genes that exhibited significant differential expression (Figure 4C). In addition, we compared gene expression changes and found that 170 genes were significantly downregulated (log2FC ≤ −1; *p* ≤ 0.05) and 388 genes were upregulated (log2FC ≥ 1; *p* ≤ 0.05) in A375R cells treated with vemurafenib and EPZ compared to A375R cells treated with vemurafenib alone (Figure 4D). In reactome pathway analysis for genes downregulated by vemurafenib and EPZ in A375R cells, we found that the most enriched pathways were associated with processes that maintain cell division (Figure 4E). Accordingly, in KEGG gene-enrichment analyses (GSEA), we received a strong depletion of genes contributing to cell cycle after treatment with vemurafenib and EPZ (Figure 4F). These results confirm, at the molecular level, that the inhibition of BRAF and EZH2 leads to a massive change in gene expression, suppressing entire signaling pathways involved in cell division and cell cycle, thus supporting our findings described above.

### 2.5. PLK1 Expression Is Mediated by BRAFi and EZH2 Inhibition

One of the most enriched pathways in the reactome pathway analyses was Polo-like kinase (PLK)-mediated events. Because Polo-like kinase 1 (PLK1) has a significant impact on cell division and cell cycle, we analyzed PLK1 expression in our NGS dataset. We found that PLK1 was highly significantly downregulated in A375R cells after treatment with vemurafenib and EPZ compared with all other conditions (Figure 5A). In Western blot analyses, we additionally showed that the PLK1 protein was similarly downregulated in A375 cells after treatment with vemurafenib alone and in combination with EPZ (Figure 5B). In contrast, vemurafenib alone reduced the PLK1 protein in A375R cells by only about 30%, whereas the combination of vemurafenib and EPZ reduced the PLK1 protein by 60% (Figure 5B). EPZ alone had no effect on the protein level of PLK1 in either A375 or A375R cells (Figure 5B). We observed similar results in additional melanoma cell lines (WM35/WM35R and WM9/WM9R) (Appendix A). Furthermore, we found that vemurafenib mediated the downregulation of PLK1 more strongly with increasing EPZ concentrations reflecting a dose–effect relationship (Appendix A).

When we analyzed our NGS dataset by GSEA, we found a strong depletion of genes associated with the PLK1 pathway for A375R cells after treatment with vemurafenib and EPZ (Figure 5C,D). To understand the link between EZH2 and PLK1, we analyzed our NGS data and found that EZH2 expression was significantly downregulated in A375R cells only after treatment with vemurafenib and EPZ, similar to PLK1 (Figure 5E). When we analyzed additional public datasets, we could confirm a correlation between EZH2 and PLK1 (Figure 5F,G). Treatment of various BRAF-mutated melanoma cell lines with the BRAF inhibitor dabrafenib (GSE98314) reduced the expression of EZH2 and PLK1 in a comparable manner (Figure 5F). In The Cancer Genome Atlas (TCGA), we found a significant correlation (r = 0.5; *p* ≤ 0.0001) between EZH2 and PLK1 expression in the cutaneous melanoma cohort (Figure 5G). These results suggest that PLK1 is an effector downstream of EZH2 that contributes to the emergence of resistance to vemurafenib.

### 2.6. PLK1 Is Associated with Tumor Progression and Poor Survival and Offers a Suitable Target for Melanoma Therapy

After demonstrating that treatment with a BRAF inhibitor leads to the suppression of PLK1 in susceptible but not in resistant cells, we reanalyzed the published dataset (GSE50509) of melanoma tumor samples from patients before therapy and at the time of tumor progression during dabrafenib therapy to determine a possible correlation between BRAFi (dabrafenib) response and PLK1 expression. We found a similar expression of PLK1 before treatment and after the emergence of resistance during BRAFi (dabrafenib) treatment (Figure 6A). In addition, we investigated the impact of PLK1 expression on the survival of melanoma patients. Using the TCGA dataset, we demonstrated that a high PLK1 expression was associated with significantly worse overall survival in melanoma patients carrying a BRAF_V600E_ mutation (Figure 6B). Because we have shown that a high PLK1 expression is associated with the occurrence of resistance and worse patient outcomes, we wondered whether PLK1 inhibition affects melanoma cell viability. To this end, we used the PLK1 inhibitor Volasertib (Vol), which has been shown to selectively block PLK1 kinase activity and function. Volasertib indeed strongly decreased the viability of both A375 (IC_50Vol_ = 6.25 nM) and A375R (IC_50Vol_ = 12.74 nM) cell lines (Figure 6C). The combination of vemurafenib and volasertib showed a synergistic effect on cell viability, resulting in an enhanced response of A375R cells to vemurafenib (Figure 6D). Similarly, when looking at apoptosis, we found that treatment with volasertib and vemurafenib in combination resulted in a great increase in apoptosis in A375R cells compared with the control, vemurafenib-treated and even volasertib-treated cells, although this increase was not significant for volasertib (Figure 6E).

Therefore, we investigated whether volasertib could serve as a suitable therapeutic strategy for melanoma associated with BRAF_V600E_ without having cytotoxic effects on normal cells. We treated normal melanocytes, normal keratinocytes, as well as A375 and A375R cells with different concentrations of volasertib for 72 h. In cell viability assays, we observed a strong induction of cell death in both melanoma cell lines, while volasertib had no effect on the viability of primary normal melanocytes and keratinocytes (Figure 6F). These results suggest that targeting PLK1 alone or in combination with BRAF inhibitors may serve as a novel therapeutic option in melanomas, particularly in melanomas with a BRAF_V600E_ mutation in which resistance to BRAFi has developed.

## 3. Discussion

We and others found that BRAF pathway inhibition by vemurafenib significantly reduced the expression of EZH2 [48,49]. In our previous work, we elaborated that EZH2 is expressed even more strongly in melanoma cells resistant to vemurafenib than in cells that respond to it [48]. Therefore, we wondered whether EZH2 could serve as a therapeutic target in melanomas resistant to BRAFi.

We now reveal that the inhibition or silencing of EZH2 contributes to the enhancement of response to vemurafenib in resistant melanoma cells, that treatment with vemurafenib and EPZ suppresses genes which are involved in the regulation of mitosis and cell-cycle progression, and finally, that the effects mediated by the inhibition of EZH2 can be explained by the downregulation of PLK1.

The improvement in the response to vemurafenib by EPZ is demonstrated by decreased cell viability, an increase in apoptosis and enhanced cell-cycle arrest of melanoma cells. In previous reports, EZH2 was already shown to contribute to melanoma progression by silencing distinct tumor suppressor genes [33,50], which results in aggressive tumor subgroups of cutaneous melanoma with higher proliferation rates and increased metastasis [33,38,44,51]. The function of EZH2, particularly in BRAF-mutated melanoma and in the emergence of resistance to targeted therapies, was incompletely understood. The coexistence of BRAF_V600E_ and EZH2 was reported to be associated with poorer overall survival and disease-free survival of patients [43]. In addition, treatment with vemurafenib in combination with the EZH2 inhibitor GSK126 has been shown to result in improved inhibitory efficacy in vitro and in vivo compared with vemurafenib alone in BRAFi-susceptible melanoma [43]. In contrast to this, we observed synergistic effects of the combined inhibition of BRAF and EZH2 only on melanoma cells resistant to vemurafenib, whereas in susceptible melanoma cells, we observed no additional benefit compared to treatment with vemurafenib alone. These discrepancies could be due to different quality in the EZH2 inhibitors (EPZ vs. GSK126) used in the studies, or due to different concentrations of inhibitors used. In contrast to these studies, our work focused on whether EZH2 contributes to the development of resistance to vemurafenib and whether it is possible to overcome this by inhibiting EZH2.

The inhibition of epigenetic regulators to improve the treatment of BRAF-mutated melanoma has also been investigated in previous works. Here, it was shown that the suppression of histone deacetylases (HDACs) in combination with BRAF inhibitors also leads to increased apoptosis and decreased tumor growth in vivo [52,53]. These results support our data, as prior deacetylation is a prerequisite for histone methylation.

EZH2 has been shown to contribute to the progression of various cancers via different mechanisms of action [54,55]. As such, EZH2 acts as an epigenetic regulator that inhibits gene expression by maintaining H3K27 trimethylation or by mediating DNA methylation. It also works as a transcriptional co-activator that induces gene expression. To understand the synergistic effects of the inhibition of EZH2 and BRAF, we performed next-generation sequencing and found major differences in gene expression, especially for A375R cells treated with a combination of vemurafenib and EPZ in comparison to the control (DMSO), EPZ and vemurafenib alone. These results are also consistent with biological assays that showed the greatest effects on apoptosis, cell cycle and cell viability following the combination of vemurafenib and the inhibition of EZH2. A comparison of differentially expressed genes between cells treated with vemurafenib and EPZ and cells treated with vemurafenib alone showed the downregulation of genes related to DNA replication, mitosis, cell division and cell cycle.

A key regulator of these processes is Polo-like kinase 1 (PLK1), a proto-oncogene whose overexpression is frequently observed in tumor cells (among others colorectal cancer, lung cancer, breast cancer and melanoma) [56,57,58,59,60,61,62,63,64]. The serine/threonine-protein kinase PLK1 triggers cell-cycle progression (G2/M transition) and supports the maturation of the centrosome and the establishment of the bipolar spindle during mitosis [65]. The inhibition or loss of PLK1 induces apoptosis in various cancer cells, including melanoma [66,67]. Our data now provide the first evidence that the whole PLK1 pathway is strongly downregulated after the inhibition of BRAF and EZH2 in resistant melanoma cells. This is supported by the correlation between the expression of EZH2 and PLK1, which we could validate in further public datasets. We confirmed this correlation both for treatment with BRAF inhibitors and without treatment. Because the transcription factor E2F1 has been shown to regulate PLK1 expression [68,69] and EZH2 has been shown to cooperate with E2F1 to stimulate gene expression [70,71], we hypothesize that this is one possible way for EZH2 to mediate PLK1 expression. Further evidence for the EZH2/E2F1-dependent expression of PLK1 is also provided by our data, showing a downregulation of EZH2 (Figure 5E) and E2F1 in A375R cells after the inhibition of EZH2 and BRAF (Appendix A).

Another possible explanation for the influence of EZH2 on PLK1 expression could be STAT3. As such, increased phosphorylation of STAT3 has been demonstrated in melanoma cells resistant to vemurafenib, while the inhibition of STAT3 leads to a better response of these cells to vemurafenib [72]. Furthermore, EZH2 has been shown to lead to increased phosphorylation and thus increased activity of STAT3 through direct methylation of the STAT3 protein [32]. Additionally, STAT3 was described to directly activate the transcription of PLK1 [73]. Thus, the inhibition of EZH2 could lead to decreased STAT3 activity, which could explain the decreased PLK1 expression.

Because we revealed that PLK1 is downregulated by vemurafenib in sensitive melanoma cells markedly stronger than in resistant cells, we hypothesize that PLK1 may be an important factor in resistance to vemurafenib. This is also supported by our in silico findings (GSE50509), which show comparable expression levels of PLK1 in progressive melanomas during treatment with BRAF inhibitors compared to the time before treatment. In addition, we found that high expressions of PLK1 lead to significantly worse overall survival (TCGA) of patients with BRAF_V600E_ mutations. Because PLK1 has already been shown to contribute to melanoma progression and poorer outcomes in general [63,66,67,74,75], we specifically focused on the relevance of PLK1 in BRAF-mutated melanoma and further investigated whether it may serve as a therapeutic target to overcome BRAFi resistance. While treatment with the PLK1 inhibitor volasertib has been shown to cause cell-cycle arrest, decreased cell viability and increased apoptosis in melanoma cells [67,75], we have found that this is also true for melanoma cells that are resistant to vemurafenib. Although PLK1 inhibitors have been investigated widely in preclinical studies, to date, monotherapy with these small molecule inhibitors, including volasertib, has not achieved satisfactory results in clinical trials [76] due to dose-limiting toxicities [77] or low intra-tumoral drug levels [78]. To overcome this issue, researchers have started to investigate treatments with PLK1 inhibitors in combination with other agents. Previous studies describe synergistic anti-tumor activity by combining the inhibition of PLK1 and NOTCH in melanoma or PLK1 [79] and MEK in NRAS-mutated melanoma [74]. When we treated resistant melanoma cells with a combination of vemurafenib and volasertib, we found synergistic effects comparable to treatment with vemurafenib and EPZ. This further supports our hypothesis that EZH2 and its downstream target, PLK1, are important factors in resistance to vemurafenib.

In summary, the combined inhibition of EZH2 and BRAF improves anti-cancer effects through enhanced cell-cycle arrest and the increased cell death of melanoma cells resistant to vemurafenib. These effects are mediated in part by the downregulation of PLK1, which promotes cancer progression by fueling cell-cycle arrest and proliferation.

Thus, the combination of BRAF inhibitors with inhibitors against EZH2 or against downstream targets of EZH2, such as PLK1, represent new therapeutic options for melanoma.

## 4. Materials and Methods

### 4.1. Cell Culture

BRAF_V600E_-mutated melanoma cell lines (A375, WM35 and WM9) were cultured in DMEM supplemented with 10% fetal calve serum (FCS) (Sigma Aldrich, Taufkirchen, Germany) and 1% penicillin-streptomycin (Sigma Aldrich, Taufkirchen, Germany). Resistant cell lines A375R, WM35R and WM9R were generated as described [48] and cultured with the continuous presence of 2 µM vemurafenib (PLX4032; LC-Laboratories, Woburn, MA, USA). Primary normal human epidermal melanocytes (NHEM) were isolated in our laboratory from juvenile foreskins and cultured in medium 254 (Thermo Fisher Scientific, Waltham, MA, USA), including human melanocyte growth supplement (HMGS) (Thermo Fisher Scientific, Waltham, MA, USA) and 1% penicillin-streptomycin (Sigma Aldrich, Taufkirchen, Germany). Keratinocytes were isolated in our laboratory from juvenile foreskins and cultured in a keratinocyte medium (Gibco, Thermo Fisher Scientific, Waltham, MA, USA), supplemented with epidermal growth factor (EGF) (Gibco, Thermo Fisher Scientific Waltham, MA, USA), bovine pituitary extract (BPE) (Gibco, Thermo Fisher Scientific Waltham, MA, USA) and 1% penicillin-streptomycin (Sigma Aldrich, Taufkirchen, Germany). All cells were incubated at 37 °C and 5% CO_2_.

### 4.2. Plasmids and Cell Transfection

Plasmids pSMP-luc (control), pSMP-EZH2_1 (plasmid # 36387), pSMP-EZH2_2 (plasmid # 36388) and pSMP-EZH2_3 (plasmid # 36389), were a gift from George Daley (Addgene, Watertown, MA, USA) [80]. A375 and A375R cells were transfected using Amaxa Cell line Nucleofector Kit V (Lonza, Basel, Switzerland) following the manufacturing instructions. Transfected cells were cultured in the presence of puromycin (Sigma Aldrich, Taufkirchen, Germany).

### 4.3. Cell Viability Assay

Cells (3000 per well) were seeded in 96-well plates. After 24 h, the cells were treated with vemurafenib, tazemetostat (EPZ-6438) (Selleck Chemicals LLC, Houston, TX, USA), vemurafenib and EPZ or volasertib (BIOZOL, Eiching, Germany) as well as volasertib and vemurafenib in combination for at least 72 h. CellTiter-Glo^®^ Luminescent Cell Viability Assay (Promega, Madison, WI, USA) was used according to the manufacturing protocol to determine the number of viable cells. Luminescence was measured by TECAN plate reader.

### 4.4. Cell Growth Assay

For growth curves, 1 × 10^5^ cells were seeded in each well of a 6-well plate, and cell numbers were determined over the indicated time periods using a TC20 cell counter (BIORAD, Hercules, CA, USA).

### 4.5. Cell-Cycle Analysis

For cell-cycle analysis, cells were trypsinized and fixed with 70% ice-cold ethanol for 30 min at 4 °C. After washing twice with PBS (Gibco, Thermo Fisher Scientific Waltham, MA, USA), 5 µL RNAse A (20–40 mg/mL) (Thermo Fisher Scientific, Waltham, MA, USA) was added and incubated for 30 min at 37 °C. After the addition of 200 µL PI (from 50 µg/mL stock solution) (Abcam, Cambridge, UK), the stained cells were analyzed on a BD Accuri C6 Plus Flow Cytometer (BD Biosciences, Franklin Lakes, NJ, USA) using FlowJo software.

### 4.6. Apoptosis Assay

Apoptosis was determined using the PE Annexin V Apoptosis Detection Kit I (BD Bioscience, Franklin Lakes, NJ, USA). Cells were trypsinized, washed with cold PBS and resuspended in a binding buffer containing 5 µL of 7-aminoactinomycin D (7-AAD) and Annexin V each. After incubation for 15 min in the dark, apoptosis was analyzed on a BD Accuri C6 Plus Flow Cytometer (BD Biosciences, Franklin Lakes, NJ, USA) using FlowJo software.

### 4.7. Western Blot Analyses

Cells were lysed by an RIPA buffer for 30 min at 4 °C. A total of 25 µg of protein extracts were resolved by SDS–PAGE and blotted to nitrocellulose membranes (Invitrogen, Thermo Fisher Scientific Waltham, MA, USA ) and probed with the following antibodies (all 1:1000): anti-EZH2 (D2C9); anti-PLK1 (208G4); anti-ERK1/2 (137F5); anti-P-ERK1/2 (Thr202/tyr204); anti-H3 (D1H2); anti-H3K27me3 (C36B11) (all Cell Signaling Technology, Leiden, The Netherlands); and anti-GAPDH (0411) (Santa Cruz, Dallas, TX, USA). Antibody incubation was performed in 5% milk (Carl Roth, Karlsruhe, Germany) or bovine serum albumin (BSA (Merck, Darmstadt, Germany)) at 4 °C overnight. For antibody detection, blots were incubated for 1 h at room temperature with m-IgGκ BP-HRP (1:5000) (Santa Cruz, Dallas, TX, USA) or anti-rabbit IgG-HRP (1:2000) (Cell Signaling Technology, Leiden, The Netherlands). Chemiluminescent detection was performed using Amersham ECL Prime (GE Healthcare, Amersham, UK).

### 4.8. Next-Generation Sequencing Analysis

To analyze the transcriptome of A375R cells treated with DMSO (control), vemurafenib, EPZ or vemurafenib and EPZ, total RNA was isolated by TRIzol RNA isolation reagents (Thermo Fisher Scientific, Waltham, MA, USA) according to the manufacturer’s instructions. RNA samples from 3 biological independent experiments were analyzed by Novogene Co, Ltd. (Cambridge, UK). Quality control, sequencing and bioinformatics were performed by Novogene as a service. The detailed description can be found in Appendix A.

Pathway analysis was performed using package enrichR (v.3.0) and R (v.4.1.2). Only terms that achieved a specific threshold of an adjusted *p*-value < 0.05 and a minimum of two overlapping genes were kept. Pathway dot plots were created using packages forcats and ggplot.

### 4.9. Datasets

The published datasets GSE98314 and GSE50509 [22] were obtained from GEO DataSets (https://www.ncbi.nlm.nih.gov/gds accessed on 22 April 2021). GSE98314 and GSE50509 gene-expression data were normalized using the cubic spline function. TCGA expression data (log2(RPM + 1) of PLK1 and EZH2 status of melanoma samples were used from The Cancer Genome Atlas (https://www.cancer.gov/tcga accessed on 22 April 2021).

### 4.10. Heatmaps and Statistics

For statistical analyses and graphical representation, Qlucore Omics Explorer and Graph Pad Prism software were used. Differential expression fold-changes and *p*-values of NGS data were analyzed using Qlucore Omics Explorer. Significantly altered genes with *p* ≤ 0.05 were selected and the Benjamini–Hochberg adjustment was used to adjust *p*-values. Adjusted *p*-values ≤ 0.05 were considered significant. To prove the statistical significands of the data, two-tailed Student’s *t*-test or the Mann–Whitney U-test was performed, depending on the presence or absence of a Gaussian distribution as evaluated by Levene’s test. For experiments with multiple groups, statistical analyses were performed using one-way ANOVA, followed by Dunnett’s comparisons test against the control group. A *p*-value of 0.05 or less was considered significant. All results are represented as the average ± standard deviation from at least three independent experiments.

## Figures and Tables

**Figure 1 ijms-24-01963-f001:**
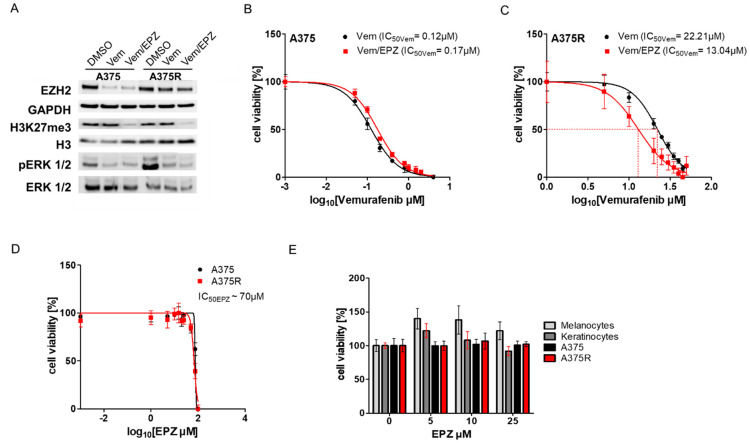
Inhibition of EZH2 improves response to vemurafenib in resistant cell lines. (**A**) Western blot of susceptible cells (A375) and cells resistant to vemurafenib (A375R), treated with DMSO (control), vemurafenib (Vem; A375 1 µM, A375R 10 µM) or vemurafenib and tazemetostat (EPZ; 5 µM), investigated EZH2, GAPDH, H3K27me3, H3, pERK 1/2 and ERK 1/2. Viability analyses of (**B**) A375 cells and (**C**) A375R cells treated with increasing concentrations of vemurafenib monotherapy (A375 0-4 µM; A375R 0-50 µM) or in combination with EPZ (5 µM). Cell viability of (**D**) A375 cells and A375R cells as well as (**E**) normal melanocytes and normal keratinocytes treated with increasing concentrations of EPZ (0-100 µM).

**Figure 2 ijms-24-01963-f002:**
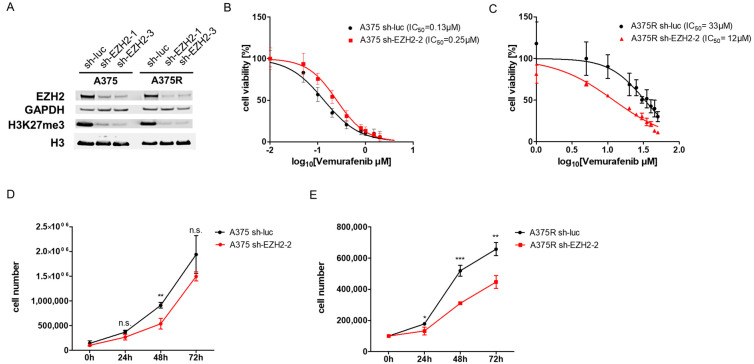
Knockdown of EZH2 improves response to vemurafenib of A375R cells. A375 cells and A375R cells were transfected with EZH2-specific shRNA constructs or control shRNAs. (**A**) Western blot of A375 cells and A375R cells, with stable knockdown of EZH2 (sh-EZH2) or control (sh-luc), examined EZH2, GAPDH, H3K27me3 and H3. Viability analysis of (**B**) A375 cells and (**C**) A375R cells, with stable knockdown of EZH2, treated with increasing concentrations of vemurafenib (Vem; A375 0-4 µM; A375R 0-50 µM). Cell proliferation of (**D**) A375 cells and (**E**) A375R cells with stable knockdown of EZH2. (* *p* < 0.05; ** *p* < 0.01; *** *p* < 0.001; n.s.: not significant).

**Figure 3 ijms-24-01963-f003:**
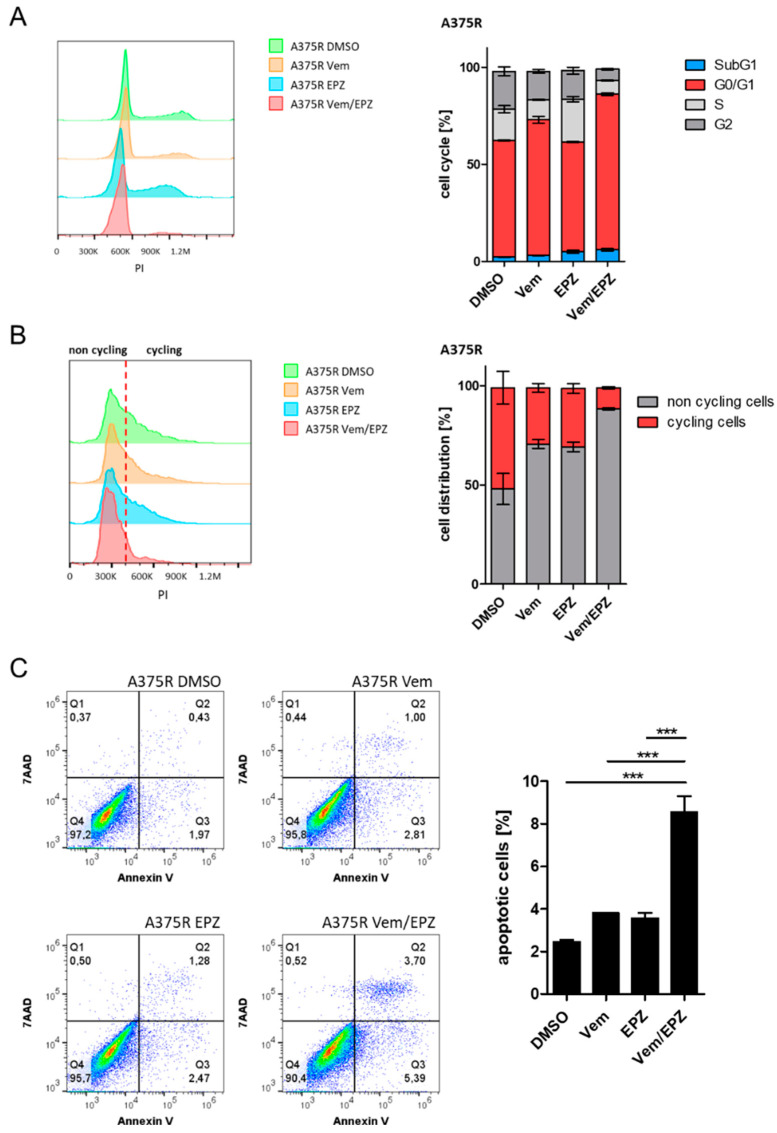
Combination of vemurafenib and EPZ enhances G0/G1 phase arrest and apoptosis in A375R cells. Cell-cycle analyses of A375R cells treated with DMSO, vemurafenib (Vem) (10 µM), EPZ (5 µM) or vemurafenib and EPZ in combination for 72 h. (**A**) Histogram and bar chart represent cell-cycle analyses of treated A375R cells. (**B**) Histogram and bar chart represent cycling and non-cycling cells after serum starvation and treatment as described above. (**C**) Analysis of apoptosis for A375R cells treated as indicated. (*** *p* < 0.001).

**Figure 4 ijms-24-01963-f004:**
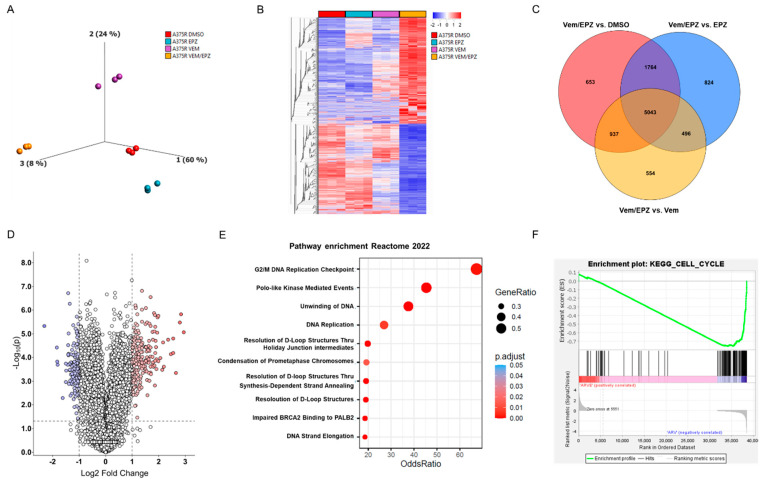
The combined treatment of A375R cells with vemurafenib and EPZ results in a significant change in the expression of cell-cycle-associated genes. (**A**) Principal component analysis (PCA) of the gene profiles of A375R cells treated with DMSO (control), EPZ (5 µM), vemurafenib (Vem) (10 µM) or vemurafenib and EPZ. (**B**) Heatmap represents significantly differentially expressed genes of A375R cells treated as described above. (**C**) Venn diagram of downregulated genes in A375R cells treated with vemurafenib and EPZ compared to all other conditions. (**D**) Volcano plot of differentially expressed genes in A375R cells treated with vemurafenib and EPZ compared to vemurafenib treatment alone. (**E**) Reactome pathway analysis for genes downregulated by vemurafenib and EPZ in A375R cells. (**F**) KEGG gene-enrichment analysis for genes downregulated by vemurafenib and EPZ in A375R cells.

**Figure 5 ijms-24-01963-f005:**
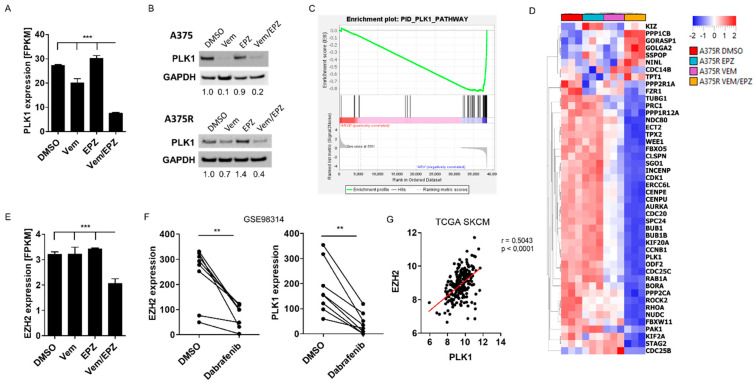
PLK1 expression is mediated by BRAFi and EZH2 signaling. (**A**) Polo-like kinase 1 (PLK1) expression (fragments per kilobase of transcript) in A375R cells treated with DMSO (control), EPZ (5 µM), vemurafenib (Vem) (10 µM) or vemurafenib and EPZ. (**B**) Western blot of A375 cells and A375R cells, treated with DMSO, vemurafenib (A375 1 µM or A375R 10 µM), EPZ (5 µM) or with vemurafenib and EPZ in combination, investigated PLK1 and GAPDH. (**C**) Enrichment plot for PLK1 pathway and (**D**) heatmap for genes downregulated by vemurafenib and EPZ in A375R cells. (**E**) EZH2 expression (fragments per kilobase of transcript) in A375R cells treated with DMSO (control), EPZ (5 µM), vemurafenib (10 µM) or vemurafenib and EPZ. (**F**) Analyses of EZH2 and PLK1 expression in eight BRAF-mutated melanoma cell lines treated with DMSO (control) or dabrafenib (GSE98314). (**G**) Correlation of EZH2 and PLK1 expression in BRAF-mutated melanoma patients (TCGA). (** *p* < 0.01; *** *p* < 0.001).

**Figure 6 ijms-24-01963-f006:**
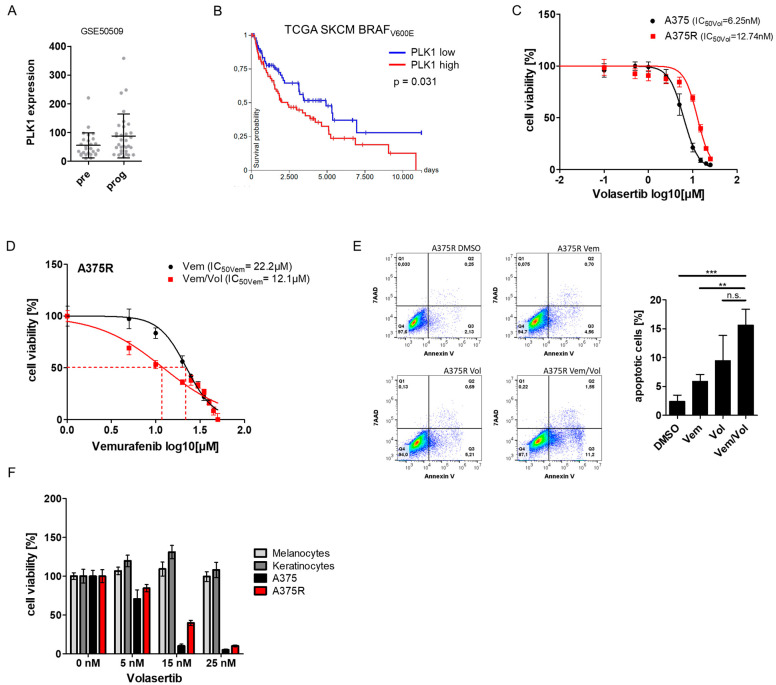
PLK1 is associated with tumor progression and poor survival and offers a suitable target for melanoma therapy. (**A**) Analysis of PLK1 expression in the dataset GSE50509 comparing untreated melanoma samples and samples at tumor progression during BRAFi treatment. (**B**) Kaplan–Meier plot of PLK1 (red = high expression, blue = low expression) in BRAF_V600E_-mutated melanoma (TCGA). (**C**) Viability assay of A375 cells and A375R cells treated with increasing concentrations of PLK1 inhibitor volasertib. (**D**) Cell viability of A375R cells treated with vemurafenib alone or vemurafenib and volasertib (10 nM). (**E**) Analysis of apoptosis for A375R cells treated with DMSO, vemurafenib (10 µM), volasertib (10 nM) or vemurafenib and volasertib. (**F**) Cell viability of A375 cells, A375R cells, as well as normal melanocytes and normal keratinocytes incubated with increasing concentrations of volasertib (0–25 nM). (** *p* < 0.01; *** *p* < 0.001; n.s.: not significant).

## Data Availability

The data presented in this study are available on request from the corresponding author.

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
