# Peer review of "Resistance to BRAF Inhibitors: EZH2 and Its Downstream Targets as Potential Therapeutic Options in Melanoma"

_ijms, 2023, doi:10.3390/ijms24031963_

Round 1

Reviewer 1 Report

This is a very useful study as there is still a great demand for therapeutics against BRAF-mutated melanoma. Combining inhibitors targeting different proteins is a promising way to address the issue.

This is a nice piece of work, data are clearly presented and adequately interpreted. 

Minor points:

1. What is the p-value of pre- and prog columns of Fig. 6A or in other words, is the difference in PLK1 gene expression statistically significant?

2. Please add headings to Fig. 6 E (similar to Fig. 3C). Otherwise is is difficult to follow the results.

3. Does Volasertib affect the PLK1 protein levels or only its kinase activity?

Author Response

Response to Reviewer 1:

This is a very useful study as there is still a great demand for therapeutics against BRAF-mutated melanoma. Combining inhibitors targeting different proteins is a promising way to address the issue. This is a nice piece of work, data are clearly presented and adequately interpreted. Thank you for your kind review of our manuscript. In the following, we will answer the questions raised.

Minor points:

  1. What is the p-value of pre- and prog columns of Fig. 6A or in other words, is the difference in PLK1 gene expression statistically significant? As described in the manuscript (line 264), the difference in PLK1 expression is not significant (p = 0.2; paired t-test). With these in vivo data, we wanted to confirm our results obtained with cell lines. showing that the expression of PLK1 is more weakly suppressed in resistant cells after BRAFi treatment than in susceptible cells.
  2. Please add headings to Fig. 6 E (similar to Fig. 3C). Otherwise is is difficult to follow the results.
    We added the headings in Figure 6E as suggested by the reviewer.
  3. Does Volasertib affect the PLK1 protein levels or only its kinase activity?
    Volasertib had no effect on PLK1 protein levels. Volasertib is a small molecule that competitively binds to the ATP-binding pocket of PLK1 protein between the NH2-terminal end and the COOH-terminal lobe of the kinase domain via two hydrogen bonds, resulting in catalytic inactivation of PLK1. Since PLK1 is phosphorylated in the G2/M state and the addition of Volasertib leads to G2/M arrest, an increase in PLK1 phosphorylation is detectable after treatment with Volasertib (please see figure below).

Reviewer 2 Report

The authors in this study obtained an interesting finding: targeting EZH2 or its downstream targets such as PLK1 in combination with BRAF inhibitors are potential novel therapeutic options in melanomas with BRAF mutations. The authors provided adequate in vitro evidences and some supporting clinical data from public datasets; and the paper is well organized and presented logically. However, the novelty of the article is limit since the relations between the biomarkers EZH2/PLK1 and melanoma are known in previous investigations. To improve the novelty and scientific values of the article, following evidences should be supplemented (1) how PLK1 is regulated by EZH2? (2) what is the behind mechanism of the synergistic therapeutic actions by targeting EZH2/PLK1 in combination with BRAF inhibitors in melanomas with BRAF mutation? Does it also work in those melanoma cases without BRAF mutation? (3) Colony formation assay data and in vivo experimental evidences are necessary to verify above in vitro conclusions; (4) A graphical abstract is necessary.  

Reviewer 3 Report

The authors assessed the role of EZH2  inhibition as a potential anti-resistant effect in melanoma treatment, following their work that documented that the BRAFV600E signaling pathway mediates the expression of EZH2. Their results revealed that the response of BRAFi-resistant melanoma cells was enhanced by co-treatment with an EZH2 inhibitor and Vemurafenib, as measured by lower viability, cell cycle arrest, and higher apoptosis. When BRAF and EZH are both inhibited, mitosis and cell cycle pathways are severely suppressed, which was linked to the downregulation of Polo-like kinase 1 (PLK1). With this work, the authors suggested a new therapeutic strategy for melanoma treatment when BRAF inhibitors are combined with EZH2 inhibitors or PLK1 inhibitors, which are targets of EZH2. Overall, the study was well-designed, and the results are fully supported, so ready for publication.

Author Response

Thank you very much for the good evaluation of our manuscript.

Round 2

Reviewer 2 Report

the authors have delivered relative reasonable response for those questoions.